# The Effect of Motivators and Barriers on Attitudes and Willingness to Consume Dairy Functional Foods in Hungary

**DOI:** 10.3390/foods13213364

**Published:** 2024-10-23

**Authors:** Mohammad Mohammad, Isaac Hyeladi Malgwi, Stefano Schiavon, Orsolya Szigeti

**Affiliations:** 1Institute of Agricultural and Food Economics, Hungarian University of Agriculture and Life Sciences (MATE), Kaposvár Campus, Guba Sándor Str. 40, H-7400 Kaposvár, Hungary; 2Department of Agronomy, Food, Natural Resources, Animals and Environment (DAFNAE), University of Padova, Viale dell’ Università 16, Legnaro, 35020 Padova, Italy; stefano.schiavon@unipd.it; 3Department of Marketing, Management and Methodology, Keleti Károly Faculty of Business and Management, Óbuda University (OE) Budapest, Tavaszmező Str. 15-17, H-1084 Budapest, Hungary; szigeti.orsolya@uni-obuda.hu

**Keywords:** healthy food, functional food, dairy, attitude, motivators, barriers, willingness

## Abstract

As the global trend for healthy eating grows, firms are emphasising dairy functional foods (DFFs). This study looks into the relationship between consumer attitudes and readiness to consume DFFs, taking into account how a healthy lifestyle might both motivate and deter people from using them. Hungary was chosen because consumer behaviour towards functional foods is under-researched in comparison to Western Europe. Data were generated using a survey questionnaire from 313 respondents. The results of Partial Least Squares (PLS) analysis revealed that consumer attitudes had a considerable influence on the desire to consume DFFs. Furthermore, the motivators and barriers had a direct influence on willingness through the consumer attitudes that serve as mediators. These findings show the need for targeting consumer attitudes and lifestyle characteristics in order to increase the market acceptance of DFFs.

## 1. Introduction

Consumer food preferences are directly related to health, especially in developing nations where chronic noncommunicable diseases are of great public health importance because of the accompanying health hazards [1,2]. Proper dietary choices are widely recognised as a crucial preventive measure against these ailments [3]. Within this context, functional foods have emerged as a category of dietary products specifically designed to enhance certain physiological processes in the body [4]. These foods include natural options, products with added beneficial components, items with removed or reduced harmful elements, and those with altered components to enhance health benefits [5,6]. Motivators such as perceived health benefits, taste, and convenience can positively affect consumption. In contrast, barriers like cost, taste preferences, and lack of awareness may hinder it. Understanding these factors is essential for improving public acceptance and consumption of functional dairy products. This research aims to provide insights for stakeholders to enhance product development and marketing strategies [5,6].

The introduction of functional foods to the European market in the mid-1990s was a key economic milestone, and their popularity in the United States increased dramatically in the early 2000s [7,8,9]. However, despite the market’s continuous growth in Europe, significant disparities in consumer attitudes and opinions remain, most likely influenced by geographical origins [10]. Some consumers struggle with accurately identifying these products, which may indicate a potential decline in interest [11]. Notably, the analysis of consumer behaviour towards functional foods, being a relatively new phenomenon, is still in its early stages [12,13]. Health is an important component in understanding consumer behaviour towards functional food, with positive attitudes and purchasing likelihood increasing when items are perceived as healthy [14,15]. The focus on health aligns with the broader idea that attitudes are crucial in shaping how information about functional foods is perceived, understood, accepted, or rejected [16]. Lifestyles, on the other hand, involve both personal and environmental influences, which further influence these attitudes [17]. The study of motivators and barriers in the consumption of dairy functional foods explores how various factors influence consumer attitudes and willingness to include these products in their diet. To conduct a consumer behaviour analysis, the theory of planned behaviour (TPB) has been widely employed, taking into account many elements to measure purchasing power and consumer intentions [18,19]. Additionally, the value–attitude–behaviour (VAB) model explores how personal and social values impact behaviour [20,21].

In contrast to previous studies that primarily focused on utilitarian and hedonic values, this research investigates both the direct and indirect effects of motivators and barriers on consumer attitudes toward functional foods. The study is distinguished by its focus on the under-researched Hungarian market, contributing valuable insights into a region that has not been extensively explored in the context of dairy functional food consumption. This research aims to explore how motivators, barriers, and perceived value impact consumer attitudes and their willingness to consume various dairy functional foods, such as lactose-free and low-fat products, probiotic yogurts, calcium-fortified milk, omega-3-enriched milk, and folic acid-fortified milk. By analysing these factors, the study offers a detailed understanding of consumer behaviour and provides insights into the drivers behind the adoption of dairy functional foods in the Hungarian market. This nuanced perspective will aid in shaping effective marketing and product development strategies.

## 2. Theoretical Framework and Hypotheses

### 2.1. Theoretical Framework

The study of attitudes and beliefs toward healthy food has gained popularity since the early 1990s [22]. The development of theoretical frameworks for assessing consumer attitudes during this period included the creation of the health belief model [23]. To evaluate attitudes and willingness to consume food, models and scales have been designed. The goal of this work is to identify a model that takes the key relationships into account. After assessing the available research and determining the key elements to focus on when examining attitudes and willingness to consume functional foods, the conceptual framework for the proposed linkages is shown in Figure 1.

### 2.2. Research Hypotheses

The theory of planned behaviour found by Ajzen [18], which is one of the fundamental concepts of human behaviour, is commonly used to analyse the attitudes and behaviours of consumers in a variety of contexts. These differences are concerned with individual motivational and barrier factors as determinants of performing health-related behaviours including the consumption of functional food [13,25,26]. According to the TPB, behavioural intention has a direct impact on behaviour. Furthermore, the following three factors influence behavioural intention: attitude towards the behaviour, subjective social standards, and perceived behavioural control. Subjective norm considers “the perceived social pressure to perform or not to perform the behaviour”, perceived behavioural control shows “the perceived ease or difficulty of performing the behaviour”, and attitude towards the behaviour refers to “the degree to which a person has a favourable or unfavourable evaluation or appraisal of the behaviour” [18]. On the other hand, this study is also based on the previous framework of the value–attitude–behaviour (VAB) model, which is a theoretical basis for understanding consumer responses [21,27,28]. In general, the VAB model is similar to the TPB model, in which the attitude and norms affect the intention to behave. That is why the measurements of motivators and barriers as values that determine the attitude of the healthy behaviour variable are used, as the goal is to ascertain their relationship with attitudes and the latter’s relationships with consumer willingness and intention.

The contributions of Downe [17] and Urala & Lähteenmäki [24] were used in the development of the model that was suggested earlier. According to the TPB by Ajzen, the attitude toward a behaviour is the degree of approval or rejection of that behaviour, where a good attitude impacts the intention to execute that behaviour positively. However, Verbeke’s [29] research confirmed that attitudes, knowledge, and control over health are the main variables in functional food acceptability. Unlike earlier research, this hypothesis encompassed attitudes and willingness to use a wide variety of dairy functional foods. Similarly, this connection was analysed in the context of a broader model in which the effect of other factors may have affected the strength of the connection. Finally, the relationship was focused on the Hungarian market. Thus, the following preliminary hypothesis posits a favourable link between dairy functional foods attitude and the willingness to consume them:

**H1.** 
*Attitudes towards DFFs influence the willingness to consume DFFs.*


In a study by Sanchez & Barrena Figueroa [14] on consumer behaviour regarding functional foods, health emerged as a key factor of interest. Products promoting health benefits tend to be more appealing to consumers [30,31], supporting the model proposed by Downes [17], which suggests that a healthy lifestyle influences attitudes toward functional food consumption. Motivators and barriers to adopting health-related behaviours can be classified into personal and environmental factors. Personal factors include motivation, health concerns, and time constraints [17], while environmental factors encompass broader influences like safety, financial constraints, social support, and stress [17]. Both personal and environmental experiences shape motivators for healthy eating and increased physical activity. Personal motivators include health awareness, perceived risks of illness, and consequences of disability or death [32,33]. Jones & Nies [34] identified environmental motivators as resource availability, social support, and awareness of disease risk [32,35], with personal drivers like energy, spiritual beliefs, and weight management also playing a role. Additionally, the motivation for health and illness prevention often drives the adoption of functional diets [36]. Verbeke [29] similarly notes that the confirmation of health benefits and having a sick relative influence functional food consumption more than sociodemographic or cognitive factors. Ultimately, a strong correlation is seen between drivers of a healthy lifestyle and perceptions of functional foods.

**H2.** 
*Motivators positively influence attitudes toward DFFs.*


Barriers to the practice of dietary or physical activity were identified in the context of personal and environmental experiences [17,31]. However, different kinds of barriers are identified by Downes [17], such as personal (lack of motivation, lack of time) and environmental (lack of social support, safety concerns, lack of resources). Moreover, personal experiences identified as negative emotional reactions [32,35], physical symptoms, and health concerns [37], a lack of desire [34,37], and a lack of time were all cited as barriers [34,37]. Access to exercise facilities, racial/cultural concerns, safety, a lack of social support [34,37], financial restrictions, and bad weather were recognised as environmental experiences that acted as barriers [35,38]. However, when it comes to the barriers, Temperini et al. [39] claimed that one of the biggest ones is the lack of knowledge and confidence about transgenic foods. As a result, they argue that good labelling and transparency are essential to overcome this obstacle to the consumption of these foods. Similarly, it is hypothesised by Abood [40] that choosing a proper diet is adversely affected by a lack of nutritional information. Likewise, several perceivable barriers, such as a lack of trust and supply limitations, have a significant influence on consumer attitudes [41,42]. That is why we can conclude that there is a negative relationship between the barriers to consuming functional food and the attitude toward them.

**H3.** 
*Barriers negatively influence attitudes toward DFFs.*


The influences on eating behaviours have been thoroughly studied in the prior work of Deshpande et al. [43]. The health belief model (HBM) is a behavioural model; nevertheless, its implementation has gotten less attention. Additionally, factors affecting eating habits have been researched. House et al. [44] examined the benefits that a healthy diet could offer to consumers. Horacek et al. [45] discovered that the factors that impacted eating habits in that order were flavour, time availability, convenience, and cost. According to what the focused group stated, they tend to serve as greater barriers to healthy eating [44]. The argument made in the literature is that the consumer lifestyle has an impact on their attitude toward the consumption of healthy foods. As an illustration, Brunner [46] evaluated the association between health and overall dietary behaviour and concluded that both health and lifestyle are excellent predictors of eating behaviour and the motivation to consume healthy foods. As a result, consumers who are more concerned about their health and are more aware of healthy lifestyles are more likely to use functional foods than consumers who are less concerned about health [6,31]. From here, we can form the following positive and negative relationships between the healthy behaviour dimensions of motivators and barriers and the willingness to consume dairy functional foods:

**H4.** 
*Motivators positively influence the willingness to consume DFFs.*


**H4a.** 
*Motivators positively influence the willingness to consume DFFs through attitude as a mediator.*


**H5.** 
*Barriers negatively influence the willingness to consume DFFs.*


The core concept of the theory of TPB and VAB was that their attitudes influenced the behaviour of consumers. However, attitude played a mediator factor between the behaviour and the antecedents [13,24,26]. In the same vein, Urala & Lähteenmäki [36] affirmed, in their study on functional foods, that attitudes modulated how information was processed, adapted, used, or refused. Nevertheless, attitude was assessed in this study from this dual viewpoint, where the cognitive and emotional antecedents of attitude to this kind of product were, respectively, the motivators and barriers to the use of dairy functional foods. Because attitudes affect how people choose their food, they can aid us in comprehending why customers make certain food choices [47]. That is why we can conclude the following hypotheses accordingly:

**H5a.** 
*Barriers negatively influence the willingness to consume DFFs through attitude as a mediator.*


## 3. Research Methods

### 3.1. Survey Design

This research is based on primary data from a questionnaire answered by a sample of Hungarian individuals during April and March of 2022. The survey was randomly distributed to adult participants residing in Hungary, and prior familiarity with dairy functional foods (DFFs) was not required. The data were collected anonymously with the consent of the respondents. The total number of valid questionnaires completed, with all questions answered, was 313. Considering the current population of Hungary with about 9.663 million, the sample size used in the present study should not be considered representative of the Hungarian population. The questionnaire covers the 4 constructs of the model proposed (attitude toward DFFs, willingness to consume DFFs, motivators, and barriers), each with its different items. It also includes questions on a series of general classification variables (gender, age, educational level, and family income). A thorough review of the existing literature was conducted to select items for the four factors of the questionnaire, resulting in a final version with 18 questions. The questionnaire was translated into Hungarian to ensure that only Hungarian participants could respond. To avoid misunderstandings due to unfamiliarity with dairy functional foods (DFFs), a brief definition and photos of the DFF products available in Hungary were included on the first page. A pre-test was conducted in April, with seven Hungarian individuals participating to evaluate the clarity of the questions. Based on feedback, some items were revised, and all questions were approved. The finalized questionnaire was then distributed online via Google Forms in mid-April 2022. Most questions used a five-point Likert scale, with responses ranging from 1 (“completely disagree”) to 5 (“completely agree”) for factors like attitudes, motivators, and barriers, and from 1 (“not willing”) to 5 (“extremely willing”) for willingness to consume DFFs. The survey, which took about 5 min to complete, was shared on social media platforms like Facebook and Instagram, with reminders to encourage participation. Over 40 days, 313 valid responses were collected from a representative sample of the Hungarian population.

### 3.2. Sample Size and Composition

The socio-demographic characteristics and background of the sample size is presented in Table 1. The total sample size consisted of 313 individuals, which is not fully representative of the Hungarian population. The sample included 41.2% males and 58.8% females. Regarding the regular consumption of dairy functional foods (DFFs), 82.4% reported consuming them, while 17.6% did not. In terms of age distribution, 51.1% were between 18 and 25 years old, 21.7% were between 26 and 30, 14.1% were between 31 and 40, 8.3% were between 41 and 50, and 4.8% were over 50. Educationally, 53% held a college/university degree, 38% had completed vocational high school, 7% had a Ph.D., 1.3% had vocational school or apprenticeship training, and 0.6% had an education level of up to 8th grade. When analysing the financial situation, 56.9% reported having enough income to live comfortably and save, 22.4% described their income as very good, allowing them to save, 18.2% had just enough to live on without saving, 1.6% sometimes struggled to make ends meet, and 1% regularly faced financial difficulties. As for place of residence, 21.1% lived in villages, 49.5% in cities, and 29.4% in the capital.

### 3.3. Statistical Analysis

SmartPLS version 3 (GmbH, Ahornstraße, Bönningstedt, Germany) was used to investigate the theoretical framework [48]. The analyses were carried out using partial least squares (PLS) and a structural equation modelling (SEM) method. SEM allows researchers to examine the structural component (path model) and measurement component (factor model) in the same model at the same time, whereas PLS helps this tool to be utilized when researchers need to work with a non-normal distribution and ignore variable changes that could cause problems with model interpretation. Moreover, when it comes to the use of measuring scales that have been used in previous surveys but are incorporated in a new model. In addition, the objective is to make predictions about important target structures to get more information regarding judgments concerning future variables. In this case, it is recommended to use PLS–SEM [49]. The measurement model is based on reflective indicators. Thus, to evaluate the reflective indicators, Cronbach’s alpha and composite reliability (CR) were employed. Furthermore, convergence validity was a measure used to evaluate the correlation between different measurements of the same construct. To assess convergence validity, the amount of loading and “average variance extracted” (AVE) were used. After testing internal consistency and convergent validity, the next step was to evaluate discriminant validity. This measures how different a construct is from other constructs in the model. In addition, structural equation modelling and model fit are critical aspects of assessing the adequacy of a proposed theoretical model. In this context, the standardized root mean square residual (SRMR) was used.

## 4. Results

### 4.1. Measurement Model: Reliability and Validity

We evaluated how each item correlates to the latent components to evaluate the reliability of the measure Table 2. According to a general guideline that was provided by Hair et al. [50], manifest variables should be kept if they have loadings that are more than 0.70. This indicates that the component is responsible for at least 50% of the variation in the manifest variable. This is performed to guarantee that significant support is provided for the reliability of the formative measurements. Thus, low-loading items (“Probiotic yoghurts” and “I am unable to afford healthy foods”) were removed due to low loading value. However, three items (“Lactose-free dairy products”, “Low-fat dairy products”, and “I have too many other things to do”) had slightly lower values (0.64, 0.68, and 0.67) but were retained as their removal did not impact Cronbach’s alpha or the AVE and CR values.

Cronbach’s alpha, composite reliability (CR), and the average variance extracted (AVE) were the metrics used in the process of determining the level of internal consistency. According to Burt [51], a high level of internal consistency is considered when both Cronbach’s alpha and the CR values are above 0.7. It is also advised to use values of at least 0.5 for the AVE [52].

The square root of the AVE was compared with the correlations among the components to determine the discriminant validity of the test. It was expected that the diagonal elements would be higher than the off-diagonal components [52]. Each formative construct’s square root of the AVE surpassed 0.7, and each was higher than the correlation between the constructs. These data imply that each concept has a stronger relationship with its measurements than with the measures of other constructs (Table 3).

### 4.2. Structural Model: Goodness of Fit Statistics

The indexes of absolute fit, or how well a model fits the data in the sample [53], is a test of the theoretical framework’s goodness of fit, and it revealed the following findings which are within the acceptable bounds. The discrepancy between the measured and the expected correlation is known as the standardized root mean square residual (SRMR). In the PLS–SEM analysis, the SRMR was employed to evaluate model fit [54]. A score of less than 0.10 (or 0.08 in a more conservative form) would indicate a satisfactory match for the data [55]. The SRMR for this model was 0.076, which indicates that a well-fit model.

### 4.3. Results of SEM

Our conceptual model results in Figure 2 show how the factors are related to each other. There is a significant relationship between the attitude toward DFFs and the willingness to consume DFFs, with a *p*-value < 0.05 (0.000) and a coefficient of 0.536; that is why the first hypothesis H1 is accepted. In the same manner, we can confirm the relationship between the motivators to consume DFFs and the attitude toward DFFs, with a *p*-value < 0.05 (0.000) and a positive coefficient of 0.487; that is why the second hypothesis H2 is accepted. Following the factor of barriers to consuming DFFs, there is also a significant influence on attitude toward DFFs, with a *p*-value < 0.05 (0.021) and a negative coefficient of −0.14, leading us to confirm and accept the third hypothesis H3.

Regarding hypotheses H4 and H5, which attempt to discover the relationship between motivators and barriers to consuming DFFs and willingness to consume DFFs, we found that there is no significant relationship between the motivators to consume DFFs and the willingness to consume DFFs due to a high *p*-value > 0.05 (0.767) and a low coefficient of −0.017. The same is observed for the relationship between the barriers to consuming DFFs and the willingness to consume DFFs, with *p*-value > 0.05 (0.725) and a lower coefficient of −0.02. That is why we have to reject both hypotheses H4 and H5.

However, the relationship between the motivators and barriers to consuming DFFs and the willingness to consume DFFs is confirmed due to the mediator role of the attitude toward consuming DFFs as a full mediator type. The significant relationship of motivators → attitude → willingness is confirmed via indirect effect, with a *p*-value < 0.05 (0.000) and a positive coefficient of 0.261. The same is observed for the relationship of barriers → attitude → willingness, which is confirmed indirectly, with a *p*-value < 0.05 (0.026) and a negative coefficient of −0.075. Therefore, we do accept H4a and H5a (Table 4).

## 5. Discussion

Many studies have investigated the relationship between attitude and willingness to consume functional foods [13,24,25,26]. However, not many studies have investigated the influence of motivators and barriers on consumer attitudes and willingness to consume functional foods. In the current study, a questionnaire for assessing the attitudes toward dairy functional foods among Hungarian consumers was developed based on the TPB and VAB models, leaning on a previous study of the motivators and barriers of health behaviour (MBHB) model.

The main body of this paper focused on the relationship between attitudes and the willingness to consume dairy functional foods. It was shown that willingness to consume dairy functional food was influenced directly by attitude, with the highest coefficient of 0.536. This relationship was tested in previous studies by various authors in other countries [13,16,24]. Attitude was influenced by two direct variables of healthy behaviour, which are the motivators and barriers to consuming dairy functional foods. However, the motivators showed a higher positive impact on attitude toward dairy functional foods, with a coefficient of 0.487, compared to barriers which showed a negative impact on attitude toward dairy functional foods, with a coefficient of −0.14. This finding aligns with prior studies that demonstrated how health-conscious individuals are more likely to have positive attitudes toward functional foods due to their perceived benefits, such as disease prevention and health maintenance. For instance, a study by Annunziata & Vecchio [56] emphasized that health-related motivators are strong predictors of a positive attitude toward functional foods, reinforcing the role of health awareness in consumer choices [8]. Furthermore, Ares et al. [57] highlighted that individuals perceiving functional foods as beneficial to their health showed a higher willingness to consume them, consistent with our findings [57]. Contrary to our hypothesis, the motivators and barriers did not have a direct impact on willingness to consume dairy functional foods. Willingness to consume dairy functional foods was indirectly influenced by motivators and barriers during the attitude intervention, which is consistent with the value–attitude–behaviour (VAB) and theory of planned behaviour (TPB) models. The results show a higher coefficient of 0.261 for the relationship of motivators–attitude–willingness than for the relationship of barriers-attitude-willingness, which was −0.075, implying that barriers may have a significant negative influence on attitude and willingness but not as much as motivators. Moreover, research by Saher et al. [58] also observed that while barriers such as mistrust or lack of knowledge about functional foods can negatively impact attitudes, these factors tend to have an indirect effect on the willingness to consume such foods, still mediated by attitudes. This further supports the idea that addressing consumers’ concerns and improving their attitudes is crucial for increasing the acceptance of functional foods.

Dairy functional foods are relatively new products, and our findings on the willingness to consume them are closely linked to the influence of consumer attitudes towards adopting this type of innovative food, as reported in several previous studies [13,24,36,59]. Incorporating dairy functional foods into one’s diet offers a novel and practical approach to self-care. The benefits of consuming these products extend beyond health improvements; as shown in Urala and Lähteenmäki’s study [16], they can also enhance mood and performance, support disease prevention, address issues related to an unbalanced diet and improve overall well-being. Additionally, functional foods make it easier to maintain a healthy lifestyle. While attitudes towards functional foods strongly influence the intention to consume various types of dairy functional products, other attitudes have product-specific effects on the willingness to use them. Europeans are notably more cautious about new food products and technologies compared to American consumers [60], with concerns about the safety and acceptance of non-traditional production methods [61]. Research by Bech-Larsen & Grunert [62] in the Nordic market also indicates that health benefits alone are insufficient to significantly increase demand for functional foods.

Many studies have highlighted the role of motivators and barriers to the use of functional food on attitude and health purchasing behaviour [13,17,31]. However, they were unable to determine the direct and indirect impacts of motivators and barriers on the attitudes and desire to consume functional foods. In our model, we demonstrated the relationship between the healthy living behaviour motivators and the barriers to consuming dairy functional foods, as well as the attitudes regarding dairy functional foods, which can influence the willingness to consume these goods in the Hungarian market.

Thus, in our conceptual model, personal factors as motivators that were identified from an individual perspective, such as seeking to live longer, controlling body weight, maintaining a healthy lifestyle, and the fear of sickness because of unhealthy behaviour showed a higher impact on attitude than the barriers such as not being motivated, not finding it enticing to consume healthy food, as well as not having enough time to care about their healthy food. This could be because the primary motivating value of functional meals is their health benefits, and customers are increasingly preferring natural products and a healthy lifestyle. This is in line with the findings of Mohammad [63] and Rozin [64], who pointed out that the preference for natural products is highly related to perceived health. However, the result can be changed according to the sample classification such as consumers who are more concerned or less concerned with health [6,31]. Furthermore, our data revealed that motivators and barriers did not directly influence the willingness to consume dairy functional foods, but rather through an interventional approach. This could be because the majority of Hungarians do not want to change their food or live a better lifestyle [65].

This study has several limitations that should be acknowledged. First, while the sample of 313 Hungarian consumers provides valuable insights, it is not fully representative of the broader population, as it is skewed toward younger, more educated individuals. This may limit the generalizability of the findings. Second, the study did not differentiate between individuals who followed a health-conscious lifestyle (HC) and those who did not (NHC). Future research could investigate whether differences exist in how motivators and barriers influence the attitudes and willingness to consume dairy functional foods between these two groups, as the literature suggests varying perspectives on the impact of these factors. Additionally, the study did not examine the effects of demographic and anthropometric characteristics, such as age, gender, education level, and income. Given the significance of these variables in shaping attitudes and behaviour towards functional foods [60,66,67], future research could benefit from a more detailed analysis of their influence. Moreover, the reliance on self-reported data may introduce bias; therefore, incorporating objective measures, such as actual purchasing behaviour, in future studies could complement the self-reported information. Finally, the findings of this study are specific to the Hungarian market and may not be generalisable to other cultural or geographical contexts.

Our recommendations for companies involved in producing healthy foods are summarised as follows. Firstly, the marketing departments should consider these findings when aiming to influence consumer attitudes towards dairy functional foods, particularly highlighting the benefits of these products, especially for those already inclined towards healthy choices. Secondly, the motivators for consuming dairy functional foods significantly impact attitudes and, in turn, the willingness to consume such products. Therefore, manufacturers, research and development departments, and marketing experts should focus their campaigns on emphasising the importance of consuming healthy foods and reinforcing the concept of a healthy lifestyle. This approach would enhance the credibility of the health claims associated with the dairy functional foods and attract a broader consumer base. Finally, to gain a strong market position, companies should aim to reduce the barriers to consumption by offering innovative purchasing methods and improving accessibility to these products. Additionally, mental stimulation techniques could be employed to increase awareness of the health benefits of functional foods, encouraging less health-conscious consumers to engage with the market. The summary of the hypothesis validation is presented in Table 5.

## 6. Conclusions

Our conceptual model builds on previous research and theories by providing empirical evidence of the relationship between attitudes and willingness to consume dairy functional foods, as well as the influence of healthy behaviour as motivators and barriers on these attitudes and willingness in Hungary. Our findings align with prior studies conducted in other countries. Four of our hypotheses were confirmed, while two were rejected. The results demonstrated a clear link between attitudes and the willingness to consume dairy functional foods. Additionally, motivators were positively correlated with attitudes towards dairy functional foods, whereas barriers were negatively correlated. However, the expected relationship between willingness to consume dairy functional foods and healthy behaviour as motivators and barriers were not confirmed. Despite this, attitudes towards dairy functional foods played a significant role as a moderator, influencing the positive relationship between motivators and willingness, as well as the negative relationship between barriers and willingness to consume dairy functional foods.

## Figures and Tables

**Figure 1 foods-13-03364-f001:**
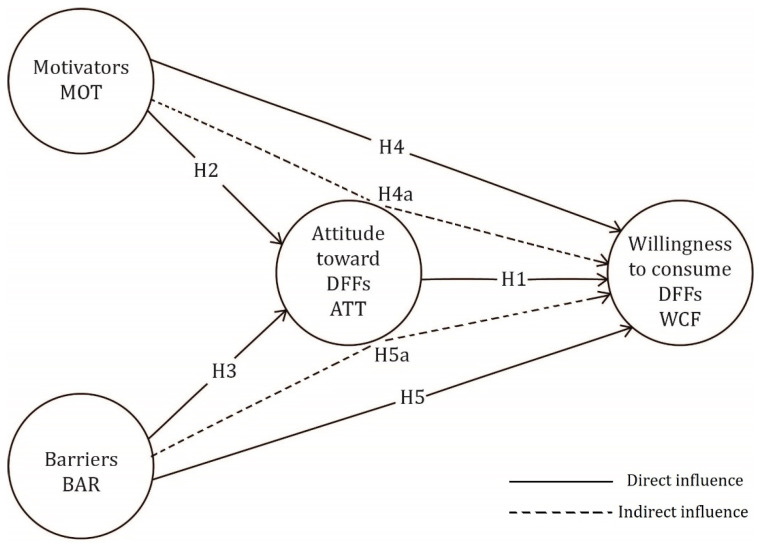
The framework of the research concept, based on Downes [17]; Urala & Lähteenmäki [24].

**Figure 2 foods-13-03364-f002:**
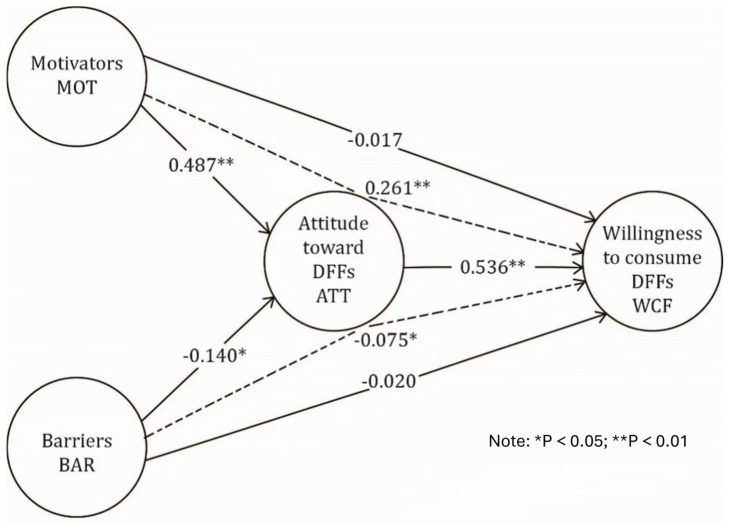
Path coefficients and *p* values of PLS-SEM using PLS–SEM.

**Table 1 foods-13-03364-t001:** Socio-demographic characteristics and background of the sample.

Gender	%	Total 313
Female	58.8	184
Male	41.2	129
Do you frequently consume DFFs?		
Yes	82.4	258
No	17.6	55
Age		
18–25	51.1	160
26–30	21.7	68
31–40	14.1	44
41–50	8.3	26
>51	4.8	15
Education level		
Maximum 8 in general	0.6	2
Vocational school/apprenticeship	1.3	4
Vocational High School/Gymnasium	38.0	119
College/University	53.0	166
PhD degree	7.0	22
How do you perceive your relative income?		
We have regular livelihood problems	1.0	3
Sometimes it’s not even enough to make a living	1.6	5
Just enough to live on, but we can no longer save	18.2	57
Enough to live, and we save from it	56.9	178
We make a very good living and we can save it	22.4	70
Where do you live?		
Village	21.1	66
City	49.5	155
Capital	29.4	92

Source: Data collected from the survey of Hungarian consumers, April 2022.

**Table 2 foods-13-03364-t002:** Constructs, items, factor loading, reliability, and validity.

Items	Factor Loadings	Sources
Attitude toward DFFs: Cronbach’s alpha: 0.88, AVE: 0.68, CR: 0.88		
DFFs help to improve my mood	0.78	[24]
My performance improves when I eat DFFs	0.86	
I can prevent disease by eating DFFs regularly	0.85	
DFFs can repair the damage caused by an unhealthy diet	0.80	
DFFs promote my well-being	0.83	
Willingness to consume DFFs: Cronbach’s alpha: 0.86, AVE: 0.65, CR: 0.86		
Lactose-free dairy products	0.64	[24]
Low-fat dairy products	0.68	
Probiotic yoghurts	*	
Milk fortified with calcium	0.90	
Enriched Milk with Omega3 EPA and DHA	0.90	
Milk fortified with folic acid	0.88	
Motivators: Cronbach’s alpha: 0.83, AVE: 0.67, CR: 0.82		
I may live longer	0.80	[17]
I want to be healthy	0.87	
I want to manage my weight	0.83	
Barriers: Cronbach’s alpha: 0.81, AVE: 0.68, CR: 0.80		
I am not motivated	0.97	[17]
I do not have someone to encourage or help me	0.80	
I have too many other things to do	0.67	
I am unable to afford healthy foods	*	

Note: AVE: Average variance extracted, CR: Composite reliability, * = Deleted item. Source: Analysis of survey responses from Hungarian consumers.

**Table 3 foods-13-03364-t003:** Discriminant validity.

	ATT	BAR	MOT	WCF
ATT	0.824			
BAR	−0.172	0.825		
MOT	0.496	−0.065	0.816	
WCF	0.531	−0.111	0.250	0.807

Source: Results from PLS–SEM analysis of survey data.

**Table 4 foods-13-03364-t004:** Hypothesis, path coefficients, and significances.

Hypothesis		Path Coefficients	T Statistics	*p* Values
H1	ATT → WCF	0.536	9.934	0.000
H2	MOT → ATT	0.487	10.123	0.000
H3	BAR → ATT	−0.140	2.304	0.021
H4	MOT → WCF	−0.017	0.296	0.767
H5	BAR → WCF	−0.020	0.351	0.725
H4a	MOT → ATT → WCF	0.261	7.001	0.000
H5a	BAR → ATT → WCF	−0.075	2.230	0.026

Source: Path analysis using PLS–SEM based on survey data.

**Table 5 foods-13-03364-t005:** Summary of hypothesis verification.

H1. *Attitudes towards DFFs influence the willingness to consume DFFs.*	Accepted
H2. *Motivators positively influence the Attitudes toward DFFs.*	Accepted
H3. *Barriers negatively influence the Attitudes toward DFFs.*	Accepted
H4. *Motivators positively influence the willingness to consume DFFs.*	Rejected
H5. *Barriers negatively influence the willingness to consume DFFs.*	Rejected
H4a. *Motivators positively influence the willingness to consume DFFs through attitude as a mediator.*	Accepted
H5a. *Barriers negatively influence the willingness to consume DFFs through attitude as a mediator.*	Accepted

Source: Results based on survey data and hypothesis testing using SEM.

## Data Availability

The original contributions presented in the study are included in the article, further inquiries can be directed to the corresponding author.

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
