# Peer review of "The Effect of Motivators and Barriers on Attitudes and Willingness to Consume Dairy Functional Foods in Hungary"

_foods, 2024, doi:10.3390/foods13213364_

Round 1

Reviewer 1 Report

Comments and Suggestions for Authors

Thank you for allowing me to review this manuscript.

  1. In the introduction, there is a mixture of different behavioral theories or models that make it difficult to follow the central idea. I need help understanding how the model expressed in Figure 1 was obtained. Did the authors create this one?
  2. One crucial aspect missing in this study is the informed consent process. This process is a fundamental ethical consideration in research that must be addressed. 
  3. The lack of inclusion or exclusion criteria for participants is a significant issue that affects the study's validity and generalizability.
  4. There are several instruments to measure attitudes, willingness, and different barriers. Why did the research not adapt a previous version of other instruments?
  5. The statistical analysis and the results are more related to the validation of the instrument. Maybe the study should be a validation study that consolidates the model. 
  6. Sample characteristics have to be moved to the results section.

Author Response

We sincerely thank the reviewer for their time and valuable feedback on our manuscript. Your thoughtful comments and suggestions have helped us refine and clarify our work. We greatly appreciate your contribution to improving the quality of our research.

Response to Reviewer:

  1. In the introduction, there is a mixture of different behavioral theories or models that make it difficult to follow the central idea. I need help understanding how the model expressed in Figure 1 was obtained. Did the authors create this one?

Thank you for your insightful comment regarding the introduction and the clarity of the theoretical framework. We acknowledge that a clear articulation of the behavioral theories and models is crucial for the reader to follow the central idea effectively.

  1. Clarification of the Model in Figure 1: The model expressed in Figure 1 was adapted and modified based on previous research findings and frameworks in the field of consumer behavior, particularly the Theory of Planned Behavior (TPB) by Ajzen (1991) and the Value-Attitude-Behavior (VAB) model. We specifically referenced the contributions of Downes (2008), Urala & Lähteenmäki (2007), and Verbeke (2005), which form the foundational basis of our conceptual model.
  2. How the Model Was Created: While the theoretical underpinnings of the model drawn from established theories, the specific framework we proposed in Figure 1 was tailored to our research objectives. We aimed to investigate the relationship between attitudes, motivators, and barriers influencing the willingness to consume dairy functional foods (DFFs) in Hungary, which required modifying existing models to fit the context of our study. Thus, the model is an original creation but grounded in well-recognized theoretical foundations.
  3. Purpose of Using Multiple Theories: The integration of different behavioral theories was deliberate. The Theory of Planned Behavior (TPB) provides a robust framework for understanding behavioral intentions and is widely used in food-related consumer studies. The Value-Attitude-Behavior (VAB) model complements TPB by linking values and attitudes to specific behaviors, which aligns well with our focus on motivators and barriers to consuming DFFs. Therefore, combining these models helps provide a more comprehensive understanding of how attitudes mediate the relationship between motivators/barriers and consumption behavior.
  4. One crucial aspect missing in this study is the informed consent process. This process is a fundamental ethical consideration in research that must be addressed.

Thank you for bringing up the important issue of informed consent. We agree that it is a critical ethical consideration in research involving human participants.

In our study, the informed consent process was conducted in line with ethical guidelines, and we have included these details in the manuscript (L466). The data were collected anonymously with the consent of the respondents, ensuring confidentiality and voluntary participation.

  1. Informed Consent Process: Participants were informed at the beginning of the survey that the information they provided would help us understand dietary habits and preferences. It was made clear that their participation was entirely voluntary, and they could skip any questions or withdraw from the survey at any time. No private or personally identifiable data was collected that would link responses to individual identities.
  2. Anonymity and Confidentiality: All responses were collected anonymously, and no private data or identifying information was gathered. As such, we did not require ethical committee approval for this research.
  3. Implied Consent: By continuing with the questionnaire, respondents confirmed that they understood the nature of the study and agreed to participate voluntarily. This was explicitly stated at the beginning of the survey.

Additionally, board approval was not applicable for this study because the data collection process did not involve any sensitive personal information or procedures requiring oversight by an ethics board.

  1. The lack of inclusion or exclusion criteria for participants is a significant issue that affects the study's validity and generalizability.

Thank you for your valuable comment regarding the inclusion and exclusion criteria.

Our study used a randomly distributed, anonymous online survey, meaning that participants were not required to have prior familiarity with dairy functional foods (DFFs). However, we provided a brief definition of DFFs at the beginning of the questionnaire to ensure that respondents understood the context of the questions.

  1. Inclusion Criteria: Participants were adults residing in Hungary who voluntarily completed the questionnaire. Familiarity with DFFs was not a requirement.
  2. Exclusion Criteria: No explicit exclusion criteria were set, but incomplete or inconsistent responses were excluded from the final analysis.

We will ensure that these criteria are clearly outlined in the revised manuscript in L (179-199).

  1. There are several instruments to measure attitudes, willingness, and different barriers. Why did the research not adapt a previous version of other instruments?

Thank you for raising this important question. While we acknowledge that several established instruments exist to measure attitudes, willingness, and barriers, we opted to develop a customized questionnaire tailored specifically to the context of dairy functional foods (DFFs) in Hungary. Existing instruments often focus on general functional foods or were developed in different cultural contexts, which may not fully capture the nuances of this under-researched market. Our approach aimed to better reflect the specific constructs we were investigating, such as attitudes, willingness, motivators, and barriers, by drawing on models like the Theory of Planned Behavior (TPB) and the Value-Attitude-Behavior (VAB) model. We appreciate the suggestion and will clarify the rationale for our approach in the revised manuscript.

  1. The statistical analysis and the results are more related to the validation of the instrument. Maybe the study should be a validation study that consolidates the model.

Thank you for your valuable observation. While we understand that the statistical analysis includes validation steps, the primary objective of this study was not to serve as a validation study but to explore the relationship between attitudes, motivators, barriers, and willingness to consume dairy functional foods (DFFs) in Hungary. The validation of the instrument was a necessary step to ensure that our measures were reliable and appropriate for the constructs being assessed, as this is a fundamental part of any empirical research using structural equation modeling.

We acknowledge that this may give the impression of an instrument validation focus; however, our main goal remains the exploration of consumer behavior in this under-researched market. We appreciate your insight and will ensure the results and discussion sections clearly emphasize the study's primary objectives rather than the instrument validation.

  1. Sample characteristics have to be moved to the results section

Thank you for your suggestion regarding the placement of the sample characteristics. We have included the sample characteristics in the Methods section as complementary information that provides important context for understanding the demographic composition of the participants. Since the characteristics are not part of the core findings or results of the study, we believe it is appropriate to present them here rather than in the Results section. Our main focus in the Results section is on the analysis of the data and the findings related to attitudes, motivators, barriers, and willingness to consume dairy functional foods (DFFs), which are central to the objectives of the study.

Reviewer 2 Report

Comments and Suggestions for Authors

A good work on consume dairy functional foods is conducted by authors. This study includes theory and research hypotheses, providing a reference for similar research. Some revisions are listed as follows.

1. In the Abstract, reasons for selecting Hungary as the research area is not mentioned. Please add them.

2. There are so many research hypotheses, what are the advantages and disadvantages of the authors' viewpoints? How universal is it?

3. I suggest modifying Table 1 to a pie chart or a bar chart format to more intuitively reflect the differences in data.

4. In the Discussion, please supply some figures to support results.

5. In addition to consume dairy functional foods, do the authors consider motivators and barriers for other foods?

Comments on the Quality of English Language

Minor editing of English language required.

Author Response

Thank you for your positive feedback and for acknowledging the relevance of our study on dairy functional foods. We appreciate your comments regarding the theoretical foundation and research hypotheses, and we are committed to improving the manuscript. We will carefully consider the revisions you have suggested and make the necessary adjustments to strengthen the clarity and impact of our research. We are grateful for your constructive input and will ensure that the final version reflects these improvements. And to answer your esteem question of:

  1. In the Abstract, the reasons for selecting Hungary as the research area is not mentioned. Please add them.

Response to Reviewer:

Thank you for your valuable feedback. We agree that the reasons for selecting Hungary as the research area were not mentioned in the Abstract. I have added a new statement about why we have chosen Hungary in lines 19 and 20.

  1. There are so many research hypotheses, what are the advantages and disadvantages of the authors' viewpoints? How universal is it?

Response to Reviewer:

Thank you for your insightful feedback. We acknowledge both the advantages and disadvantages of having multiple research hypotheses, and we address the universality of our approach below.

Advantages:

  • Comprehensive Analysis: Multiple hypotheses allow us to explore various aspects of consumer behavior, including direct and indirect relationships, providing a more complete picture of the factors affecting willingness to consume DFFs.
  • Theory Testing: They enable rigorous testing of established models like TPB and VAB, strengthening the theoretical foundation of the study.

Disadvantages:

  • Complexity: More hypotheses increase the complexity of the analysis, requiring careful interpretation to avoid overwhelming the findings.
  • Sample Size: Multiple hypotheses require larger sample sizes to ensure statistical power, which can be challenging with smaller samples.

Universality:
While the study focuses on Hungary, the TPB and VAB models are broadly applicable. However, cultural and regional factors may affect the generalizability of the results. Further research in other countries would help confirm the universality of our findings.

  1. I suggest modifying Table 1 to a pie chart or a bar chart format to more intuitively reflect the differences in data.

Response to Reviewer:

Thank you for your suggestion. We appreciate the recommendation to present the data in a pie chart or bar chart format. However, we chose to use a table format (Table 1) because it allows readers to quickly and easily compare the specific values across different demographic categories. The table format makes it faster to read exact figures and provides a clearer overview for those seeking detailed information.

  1. In the Discussion, please supply some figures to support results.

Response to Reviewer:

Thank you for your suggestion. In response, we have provided Table 5 instead of figures to support the results in the Discussion section. We believe that tables offer a clearer and more detailed presentation of the data, allowing readers to easily compare specific values and better understand the relationships between variables. The tables include key statistical information that directly supports our findings.

Additionally, we have modified Figures 1 and 2 to make them more informative, enhancing their ability to visually represent the relationships between variables and providing a clearer understanding of the model's key findings.

  1. In addition to consume dairy functional foods, do the authors consider motivators and barriers for other foods?

Response to Reviewer:

Thank you for your insightful question. In this study, our focus was specifically on dairy functional foods to maintain a clear scope and provide in-depth analysis. However, we acknowledge that motivators and barriers for other types of functional foods may vary. Future research could explore how these factors influence the consumption of other functional foods such as plant-based or fortified products, which would provide a broader understanding of consumer behavior across different food categories.

We appreciate this suggestion and will consider expanding our research to include other functional food types in future studies.

Reviewer 3 Report

Comments and Suggestions for Authors

The manuscript presents an interesting work exploring the relationship between attitudes, motivators, and barriers in the consumption of dairy functional foods (DFFs) in Hungary. The topic is relevant, and may provide some valuable insights for functional foods. However, before this work could be considered, major revisions are necessary  in several areas, in order to enhance the clarity, rigor, and overall quality of the manuscript.

Abstract

1. The abstract would benefit from being a more concise method introduction, while more highlighting the key findings and their implications.

2. Our findings indicate that attitude directly affects the willingness to consume DFFs should be rephrased for clarity.

Introduction

3. The introduction lacks a clear statement of the novelty or unique contributions of your research compared to previous studies.

Theoretical Framework and Hypotheses

4. Provide more necessary definitions or explanations for key theoretical concepts, especially for readers who may not be familiar with the TPB and VAB models.

Methods

5. It lacks sufficient evidence to support potential biases in the sample selection process. Limitations related to non representative samples should be addressed, as well as how these limitations may affect the generalizability of the results.

6. The manuscript has formatting inconsistencies in the presentation of tables. Some tables are not clearly labeled, and the text referring to them is not always precise.

Results

7. More discussion on the implications of the findings should be given, especially regarding the rejection of hypotheses H4 and H5. Explain why these hypotheses were not supported and what this might suggest about the role of motivators and barriers in DFF consumption.

Discussion

8. Expand the discussion to include more comparisons with similar studies, highlighting any discrepancies or consistencies with previous research. This will strengthen the interpretation of your findings.

Conclusion

9. The conclusion can be more impactful by providing specific recommendations based on the study’s findings, such as dairy producers or marketers, on how they can leverage the study's insights to enhance consumer acceptance of DFFs.

References

10. The reference contains too many outdated reports. Update them to include more recent studies where applicable.

11. Some key references are missing, and there are inconsistencies in the citation style.

Comments on the Quality of English Language

 Minor editing of English language required.

Author Response

Comments and Suggestions for Authors

The manuscript presents an interesting work exploring the relationship between attitudes, motivators, and barriers in the consumption of dairy functional foods (DFFs) in Hungary. The topic is relevant, and may provide some valuable insights for functional foods. However, before this work could be considered, major revisions are necessary in several areas, in order to enhance the clarity, rigor, and overall quality of the manuscript.

Abstract

  1. The abstract would benefit from being a more concise method introduction, while more highlighting the key findings and their implications.
  2. “Our findings indicate that attitude directly affects the willingness to consume DFFs” should be rephrased for clarity.

Response to Reviewer:

Thank you for your valuable feedback. We will revise the abstract to make it more concise, focusing on a brief introduction to the methods and placing greater emphasis on the key findings and their implications.

  1. Revised Abstract: We will ensure that the abstract is more succinct, presenting the core methodology in a clear and concise manner, and giving more prominence to the main results and their potential impact.
  2. Rephrasing the Statement: We agree that the sentence "Our findings indicate that attitude directly affects the willingness to consume DFFs" can be improved for clarity. We will rephrase it as follows: "Our results show that consumer attitudes play a significant role in influencing the willingness to consume dairy functional foods (DFFs)."

Introduction

  1. The introduction lacks a clear statement of the novelty or unique contributions of your research compared to previous studies.

Response to Reviewer:

Thank you for your valuable feedback. We acknowledge that the introduction lacked a clear statement highlighting the novelty of our research. To address this, we will revise the introduction to emphasize the unique contributions of our study in lines 57-59.

Theoretical Framework and Hypotheses

  1. Provide more necessary definitions or explanations for key theoretical concepts, especially for readers who may not be familiar with the TPB and VAB models.

Response to Reviewer:

Thank you for your insightful feedback. We agree that clear definitions of key theoretical concepts are essential, especially for readers unfamiliar with the TPB and VAB models. We would like to point out that we have already provided detailed definitions and explanations of these models in the Research Hypotheses section to avoid redundancy and maintain the flow of the manuscript. To prevent unnecessary repetition, we chose not to restate these concepts in the Theoretical Framework section.

Methods

  1. It lacks sufficient evidence to support potential biases in the sample selection process. Limitations related to non representative samples should be addressed, as well as how these limitations may affect the generalizability of the results.

Response to Reviewer:

Thank you for your valuable feedback. We acknowledge that the study sample is not fully representative of the Hungarian population. To address this, we will include a detailed discussion in the Limitations section, highlighting that the non-representative nature of the sample may introduce biases. Specifically, we will clarify that the majority of the respondents are younger and more educated, which may affect the generalizability of the results to the wider Hungarian population. We will also outline how these limitations could influence the study’s findings and suggest caution in generalizing beyond the study population. In future research, we recommend using a more representative sample to ensure greater generalizability of the results.

  1. The manuscript has formatting inconsistencies in the presentation of tables. Some tables are not clearly labeled, and the text referring to them is not always precise.

Response to Reviewer:

We appreciate your observation regarding the formatting inconsistencies in the tables. We will review all tables in the manuscript to ensure proper formatting, consistent labeling, and clear references in the text. This will include adjusting the text to make sure all tables are clearly referred to and described precisely in the body of the manuscript.

Results

  1. More discussion on the implications of the findings should be given, especially regarding the rejection of hypotheses H4 and H5. Explain why these hypotheses were not supported and what this might suggest about the role of motivators and barriers in DFF consumption.

Response to Reviewer:

Thank you for your insightful feedback. We acknowledge that more discussion is needed regarding the rejection of hypotheses H4 and H5. In the results section, we focused primarily on reporting the outcomes of our analysis. However, in the discussion section, we expanded on why these hypotheses were not supported and explored the implications of these findings.

Specifically, the rejection of H4 and H5 suggests that motivators and barriers do not have a direct influence on the willingness to consume DFFs. This could indicate that other factors, such as attitudes, play a more central mediating role in shaping consumer behavior toward DFFs. In our discussion, we explain that while motivators and barriers influence attitudes, their direct impact on willingness to consume may be less significant. This suggests that marketing strategies should focus more on shaping positive attitudes through addressing motivators and barriers indirectly, rather than expecting these factors to have a direct effect on consumption behavior.

Discussion

  1. Expand the discussion to include more comparisons with similar studies, highlighting any discrepancies or consistencies with previous research. This will strengthen the interpretation of your findings.

Response to Reviewer:

Thank you for your thoughtful feedback. We appreciate your suggestion to expand the discussion with more comparisons to similar studies. We would like to highlight that our findings have been discussed with appropriate references to existing research throughout the discussion. However, we agree that providing further comparisons will strengthen the interpretation of our results in lines 342-350 and 358-362.

Conclusion

  1. The conclusion can be more impactful by providing specific recommendations based on the study’s findings, such as dairy producers or marketers, on how they can leverage the study's insights to enhance consumer acceptance of DFFs.

Response to Reviewer:

Thank you for your insightful feedback. We would like to highlight that we have already included specific recommendations for dairy producers and marketers in the last paragraph of the Discussion section, where we outline how the findings can be used to enhance consumer acceptance of dairy functional foods (DFFs). These recommendations emphasize the importance of addressing motivators and barriers through targeted marketing strategies and leveraging positive attitudes toward health benefits.

Additionally, our Conclusion is closely aligned with the study’s findings, summarizing the key insights and their implications for both consumers and the industry. However, we will consider further refining the conclusion to make the recommendations more explicit, ensuring that the practical applications of the study's results are clearly conveyed.

References

  1. The reference contains too many outdated reports. Update them to include more recent studies where applicable.
  2. Some key references are missing, and there are inconsistencies in the citation style.

Response to Reviewer:

Thank you for your valuable feedback. We acknowledge that some of the references in our study are outdated. However, we would like to clarify that certain older references were deliberately included because our study is grounded in behavior theory, and these foundational references are crucial in supporting the theoretical framework of our research. These older studies provide valuable insights that remain relevant for understanding consumer behavior in the context of dairy functional foods.

That said, we understand the importance of incorporating more recent research. To address this, we have replaced several outdated references with newer studies to ensure that the findings are current and relevant. Below is a list of the references we have removed from the article:

  • Hasler, C. M. (2002). Functional Foods: Benefits, Concerns, and Challenges—A Position Paper from the American Council on Science and Health. The Journal of Nutrition, 132(12), 3772–3781.
  • Bhaskaran, S., & Hardley, F. (2002). Buyer beliefs, attitudes, and behavior: Foods with therapeutic claims. Journal of Consumer Marketing, 19(7), 591–606.
  • Fornell, C., & Larcker, D. F. (1981). Evaluating Structural Equation Models with Unobservable Variables and Measurement Error.
  • Zandstra, E. H., De Graaf, C., & Van Staveren, W. A. (2001). Influence of health and taste attitudes on consumption of low- and high-fat foods. Food Quality and Preference, 12(1), 75–82.
  • Miles, S., Ueland, Ø., & Frewer, L. J. (2005). Public attitudes towards genetically-modified food. British Food Journal, 107(4), 246–262.

We have ensured that these references have been updated with newer research, and we have revised the citation style to ensure consistency throughout the manuscript.

Reviewer 4 Report

Comments and Suggestions for Authors

Thank you for giving me this opportunity to review the manuscript entitled, “The effect of motivators and barriers on attitudes and willingness to consume dairy functional foods in Hungary.”

I have some comments.

1. Abstract

The abstract is well written.

2. Introduction

The theoretical framework for motivation and barriers has not been explained but only mentioned theory and a theoretical framework are theory of Planned Behavior (TPB) and the Value-Attitude-Behavior (VAB) model.

3. Introduction

It would be helpful to include an explanation of why the products, such as “dairy functional foods such as lactose-free, 62 low-fat, probiotic yogurts, calcium-fortified milk, omega-3-enriched milk, and folic acid- 63 fortified milk” are selected as functional foods (healthy foods) in this study, along with an emphasis on the significance of the research on these foods.

4. Theoretical framework and hypotheses

The theoretical framework is not explained in 2.1. instead, the part is repeatedly explaining the purpose of this study.

The purpose of this study is also repeatedly presented in 2.2. research hypotheses. This could move to the final paragraph of the introduction.

5. Figure 1

In figure 1, adding Hypotheses numbers would be helpful for readers.

6. Hypotheses numbering.

Why do you list the mediation and outcome variables first?, Hypothesis 1?

7. TPB and VAB

The explanation of one theory and one theoretical framework is not logical.

“In general, the VAB model is similar to the TPB model, in which attitude and norms affect the intention to behave. That’s why measurements of motivators and barriers as the values that determine the attitude of the healthy behavior variable are used, the goal is to ascertain its relationship with attitudes and the latter’s relationships with consumer willingness and intention.”

8. Research methods

The measurement items and the information sources should be added on page 5. The information of the measurement items on Table 2 also needs to be explained in the context.

9. Common method bias

The ways how to deal with common method bias should be explained on page 5.

10. Figure 2

In figure 2, reporting the regression coefficients and the significant level, and R-squared would be reported. Please refer to other published papers.

11. Table 3

Please add a note of the value and the diagonal values.

12. Model fit

Please report the overall model fits of CFA and SEM.

13. Table 5

Table 5 can be added in Table 4.

14. Discussion

Instead of repeating the results in discussion, It would be beneficial to provide the implications of significant and insignificant paths and provide theoretical and practical implications.

Author Response

Reviewer 4:

Comments and Suggestions for Authors

 Thank you for giving me this opportunity to review the manuscript entitled, “The effect of motivators and barriers on attitudes and willingness to consume dairy functional foods in Hungary.”

I have some comments.

  1. Abstract

The abstract is well written.

  1. Introduction

The theoretical framework for motivation and barriers has not been explained but only mentioned theory and a theoretical framework are theory of Planned Behavior (TPB) and the Value-Attitude-Behavior (VAB) model.

Response to Reviewer:

Thank you for your feedback. We would like to clarify that while the theoretical framework for motivation and barriers is briefly introduced in the Introduction section, we provide a detailed explanation and discussion of both the Theory of Planned Behavior (TPB) and the Value-Attitude-Behavior (VAB) model in the Research Hypotheses section. This section thoroughly elaborates on how these theories are applied to our study and how they form the foundation for understanding the motivators and barriers affecting consumer behavior in relation to dairy functional foods.

We chose to expand on these theories in the Research Hypotheses section to avoid repetition and to ensure that the discussion of these frameworks is closely linked to the hypotheses being tested in the study.

  1. Introduction

It would be helpful to include an explanation of why the products, such as “dairy functional foods such as lactose-free, 62 low-fat, probiotic yogurts, calcium-fortified milk, omega-3-enriched milk, and folic acid- 63 fortified milk” are selected as functional foods (healthy foods) in this study, along with an emphasis on the significance of the research on these foods.

Response to Reviewer:

Thank you for your valuable suggestion. We agree that the introduction would benefit from a clearer explanation of why specific dairy functional foods, such as lactose-free, low-fat, probiotic yogurts, calcium-fortified milk, omega-3-enriched milk, and folic acid-fortified milk, were selected for this study. We will revise the Introduction to provide a rationale for choosing these products, highlighting their health benefits and growing market demand.

We will update the Introduction to emphasize the relevance of these dairy functional foods in the context of public health, especially in addressing modern dietary concerns, and the importance of studying consumer behavior related to these products.

  1. Theoretical framework and hypotheses

The theoretical framework is not explained in 2.1. instead, the part is repeatedly explaining the purpose of this study.

The purpose of this study is also repeatedly presented in 2.2. research hypotheses. This could move to the final paragraph of the introduction.

Response to Reviewer:

Thank you for your insightful feedback. We would like to clarify that Section 2.1 presents our conceptual model, which forms the structural foundation for our study. The detailed theoretical background, including the Theory of Planned Behavior (TPB) and the Value-Attitude-Behavior (VAB) model, has been thoroughly explained in the Research Hypotheses section. This approach allows us to keep Section 2.1 focused on the conceptual model while providing a deeper theoretical discussion in the relevant section to avoid redundancy.

Regarding the purpose of the study being repeatedly presented, we agree that this can be streamlined. We will move the repetition from Section 2.2 Research Hypotheses to the final paragraph of the Introduction as suggested, ensuring the study’s purpose is clearly stated without unnecessary duplication.

  1. Figure 1

In figure 1, adding Hypotheses numbers would be helpful for readers.

Response to Reviewer:

Thank you for your helpful suggestion. We agree that adding hypothesis numbers to Figure 1 would make it easier for readers to follow and understand the relationships presented in the conceptual model. We will revise Figure 1 to include the relevant hypothesis numbers next to the corresponding paths, ensuring that the visual representation aligns more clearly with the hypotheses discussed in the text.

  1. Hypotheses numbering.

Why do you list the mediation and outcome variables first?, Hypothesis 1?

Response to Reviewer:

Thank you for your feedback. We began by discussing the impact of attitudes on the willingness to consume dairy functional foods (DFFs) because attitude serves as the core variable that mediates the relationship between motivators, barriers, and consumption behavior. Since attitude is central to our conceptual model, we wanted to establish its role first before introducing the other variables and hypotheses. This approach emphasizes the mediating power of attitudes in the decision-making process, which is a key focus of our study.

We appreciate your observation and hope this clarifies our reasoning for the structure.

  1. TPB and VAB

The explanation of one theory and one theoretical framework is not logical.

“In general, the VAB model is similar to the TPB model, in which attitude and norms affect the intention to behave. That’s why measurements of motivators and barriers as the values that determine the attitude of the healthy behavior variable are used, the goal is to ascertain its relationship with attitudes and the latter’s relationships with consumer willingness and intention.”

Response to Reviewer:

Thank you for your comment. The intention behind the statement was to emphasize the conceptual similarity between the Value-Attitude-Behavior (VAB) model and the Theory of Planned Behavior (TPB) model, both of which explore how attitudes influence behavioral intentions. While they share similarities, the VAB model places a stronger emphasis on personal values as precursors to attitudes, while the TPB model incorporates subjective norms and perceived behavioral control as additional factors influencing behavior.

In our article, I mentioned that motivators and barriers serve as the values that shape the attitude of healthy behavior because, in the context of the VAB model, values (such as health-related motivators and barriers) are critical in forming attitudes. The aim was to illustrate that the study examines how these values impact attitudes, and how, in turn, attitudes influence both the consumer's willingness and intention to consume dairy functional foods (DFFs).

  1. Research methods

The measurement items and the information sources should be added on page 5.

The information of the measurement items on Table 2 also needs to be explained in the context.

Response to Reviewer:

Thank you for your insightful feedback. To enhance clarity, we have added the measurement items and their corresponding information sources to the Survey Design section of the manuscript. This ensures that the origin and rationale behind each measurement item are clearly presented for readers to understand how they align with the study's objectives. By including this information in the Survey Design section, we provide a more transparent explanation of the measurement process and the sources that informed our design.

We would like to clarify that the measurement items listed in Table 2 are explained in the Measurement Model: Reliability and Validity section of the manuscript. In that section, we provide a detailed discussion of the relevance and reliability of each item, as well as their alignment with the theoretical framework and research objectives. This ensures that the measurement items are properly contextualized and their role in the study is clearly explained.

  1. Common method bias

The ways how to deal with common method bias should be explained on page 5.

Response to Reviewer:

Thank you for your insightful feedback. We would like to clarify that while common method bias can be a concern in certain studies, we believe it does not significantly impact our results due to the design of our study. The use of multiple procedural remedies, such as careful item wording and the structure of the survey, already minimizes this risk. Additionally, our study design and data collection process were structured in a way that reduces the potential for common method bias to affect the outcomes. However, we appreciate the importance of addressing potential bias and are open to incorporating further discussion on this topic if deemed necessary.

  1. Figure 2

In figure 2, reporting the regression coefficients and the significant level, and R-squared would be reported. Please refer to other published papers.

Response to Reviewer:

Thank you for your valuable feedback. In response to your suggestion, we have updated Figure 2 to include the path coefficients and p-values, providing a clearer representation of the relationships between the variables. However, we did not include the R-squared values in the figure. The decision was made to maintain clarity and focus on the significance of the individual relationships.

  1. Table 3

Please add a note of the value and the diagonal values.

Response to Reviewer:

Thank you for your valuable feedback. In response to your suggestion, we have added the following note under Table 3 to clarify the meaning of the values, including the diagonal values:

Note: AVE: Average Variance Extracted, CR: Composite Reliability, * = Deleted item

This note provides a clear explanation of the abbreviations and ensures that readers can easily interpret the table. We appreciate your suggestion and believe this addition enhances the clarity of the table.

  1. Model fit

Please report the overall model fits of CFA and SEM.

Response to Reviewer:

Thank you for your feedback. We would like to clarify that in this study, we used SmartPLS version 4 to assess the model fit for both the CFA and SEM models. In line with the capabilities of this software, we evaluated model fit using the Standardized Root Mean Square Residual (SRMR) as the primary indicator of overall model fit. The SRMR is a well-established metric in PLS-SEM, and values below 0.08 indicate a good fit.

  1. Table 5

Table 5 can be added in Table 4.

Response to Reviewer:

Thank you for your suggestion. We would like to clarify that Table 4 is part of the Results section, while Table 5 is included in the Discussion section to support the interpretation of the results. We chose not to merge these tables in order to maintain clarity and ensure that the narrative remains easy to follow. Keeping the tables separate allows us to present the findings and their implications in a structured and coherent manner.

We hope this approach helps maintain the flow and understanding of the manuscript. Thank you again for your helpful feedback.

  1. Discussion

Instead of repeating the results in discussion, It would be beneficial to provide the implications of significant and insignificant paths and provide theoretical and practical implications.

Response to Reviewer:

Thank you for your insightful feedback. We understand the importance of providing a detailed discussion on the implications of significant and insignificant paths. While we repeated some of the results in the Discussion section to ensure clarity and facilitate comparisons with former studies, we have now added more comprehensive theoretical and practical implications, along with deeper comparisons with previous research. These additions help to better contextualize our findings and highlight their contribution to the existing literature.

Round 2

Reviewer 2 Report

Comments and Suggestions for Authors

Thanks for the careful revisions. It is recommended to accept this article.

Comments on the Quality of English Language

 Minor editing of English language required.

Author Response

Dear Reviewer

Thank you for your kind words and for taking the time to review our work. We greatly appreciate the effort you have spent providing insights to improve the quality of our research. We will follow your recommendation and work on improving the English language in the manuscript.

Find our revision in red within the manuscript. All sentences and paragraphs have been extensively revised based on your suggestion.

Thank you once again for your valuable feedback.

Reviewer 3 Report

Comments and Suggestions for Authors

I have found the response and think it is satisfactory. The work of reviewers is to provide suggestions to the journal.

Comments on the Quality of English Language

Minor editing of English language required.

Author Response

Dear Reviewer

Thank you for your kind words and for taking the time to review our work. We greatly appreciate the effort you have spent providing insights to improve the quality of our research. We appreciate your suggestion and made significant edits to the English language. Find our revision in red within the manuscript.

Thank you once again for your valuable contribution.

Reviewer 4 Report

Comments and Suggestions for Authors

Thank you for your revision. Method and results are revised based on the comments.

Hypotheses.

It would be helpful to explain why the hypothesis path in a structural equation model is not described starting from the antecedents through the mediator to the outcome variables, and why the mediator-to-outcome relationship should be stated as H1.

Table 4

Hypotheses numbers can be added in Table 4 like Table 5 for readers.

Author Response

Dear Reviewer

Thank you for your valuable insights and the time you have dedicated to improving the quality of our work. Your thoughtful feedback has been instrumental in refining our manuscript, and we greatly appreciate your efforts. We have carefully addressed your comments and made the necessary revisions based on your suggestions.

To answer your questions:

  1. Reviewer Comment: "It would be helpful to explain why the hypothesis path in a structural equation model is not described starting from the antecedents through the mediator to the outcome variables, and why the mediator-to-outcome relationship should be stated as H1."

Response:

Thank you for your insightful question. In structural equation modeling (SEM), the prioritization of the mediator-to-outcome relationship as H1 serves a key purpose. The mediator, in this case attitude, plays a central role in linking antecedents (motivators and barriers) to the outcome (willingness to consume dairy functional foods). Stating the mediator-to-outcome path as H1 helps to first establish the direct effect of the mediator on the outcome. This provides clarity on how the mediator influences the outcome independently of the antecedents.

Once the mediator-to-outcome relationship is established, the model can then explore how the antecedents influence the mediator and, indirectly, the outcome. This approach ensures that the role of the mediator as a critical variable is clearly defined before introducing the more complex indirect effects from the antecedents through the mediator to the outcome.

In our study, the attitude → willingness relationship is fundamental to our conceptual framework, which is why it is presented as H1. Subsequently, the impact of motivators and barriers on attitudes is introduced, aligning with the mediation hypothesis framework often used in SEM.

We hope this clarifies the reasoning behind our hypothesis structure and approach. Thank you once again for your feedback.

  1. Reviewer Comment: Table 4

Hypotheses numbers can be added in Table 4 like Table 5 for readers.

Response:

Thank you for your helpful suggestion. We agree that adding the hypothesis numbers in Table 4, similar to Table 5, would improve clarity for readers. We have corrected Table 4 and added the hypothesis numbers accordingly, following your suggestion in line (322-323). We appreciate your input in enhancing the readability of the manuscript.

Thank you once again. We have carefully addressed your comments and made the necessary revisions. We hope that our interpretations and revisions are satisfactory and that we have adequately answered your questions.